# Correlation between Feeding Behaviors and Retinal Photoreceptor Cells of Largemouth Bass, *Micropterus salmoides*, in Korea

**Jae Goo Kim** [1,2], **Su Hwan Kim** [3], **Jong Young Park** [1] and **Su-Hyang Yoo** [3,*]

[1] Department of Biological Science, College of Natural Sciences, Jeonbuk National University, Jeonju 54896, Korea; jgkim0909@jbnu.ac.kr (J.G.K.); park7877@jbnu.ac.kr (J.Y.P.)
[2] Alpha Research Ecology Institute, Gunsan 54151, Korea
[3] Division of Ecological Safety Research, National Institute of Ecology (NIE), Seocheon 33657, Korea; ksh0814@nie.re.kr
* Correspondence: hyang77@nie.re.kr; Tel.: +82-42-950-8530

**Abstract:** The largemouth bass (*Micropterus salmoides*), a food resource in Korea, is a highly voracious predator, designated as an invasive species. It is both diurnal and nocturnal, with high adaptability and reproducibility. Since the predation characteristics are associated with sensitive dynamic visual acuity, we examined the ocular structure of the largemouth bass to investigate the association between photoreceptor cells and feeding behavior. Furthermore, the comparison of the prey-tracking ability of *M. salmoides* with that of other predatory fish (*Coreoperca herzi* and *Lepomis macrochirus*) with similar ecological characteristics revealed the structure and arrangement of photoreceptor cells, typical of a predatory fish in *M. salmoides*. The double and single cone cells in the retina were present in the regular mosaic patterns of the quadrilateral units, with four double cone cells surrounding a single cone cell. The photoreceptor cells, i.e., the rod, single, and double cone cells of *M. salmoides*, were smaller ($2.3 \pm 0.2$, $3.82 \pm 0.2$, and $7.5 \pm 0.2$ μm, respectively) than those of other species ($3.1 \pm 0.24$, $6.6 \pm 0.5$, and $11.3 \pm 0.4$ μm in diameter, respectively, in *C. herzi*). A smaller diameter of cone cells allows for a high-density arrangement of visual cells, possibly affecting the dynamic visual acuity for prey tracking.

**Keywords:** largemouth bass; feeding behavior; photoreceptor cell; cone cell; rod cell; mosaic model





## 1. Introduction

The largemouth bass (*Micropterus salmoides*) is an invasive species that was introduced to Korea as a food resource in 1973. Largemouth bass mainly inhabits the freshwater area of a ponds and slow-moving streams. It is ranked as a top predator, due to its massive consumption of aquatic invertebrate animals, as well as young and adult fish, and the absence of a natural enemy [1–3]. Combined with its highly developed lateral line and lightning-fast speed, the photoreceptor system is responsible for the high rank of this species in the food chain [4]. The largemouth bass is a diurnal species that feeds during the day, although predation pressure appears high when preys gather in the morning and evening [1,2]. Consequently, this species rapidly became widely distributed throughout freshwater systems, and its strong predation and competition has resulted in the decline and displacement of the native species [2,5–7]. Finally, in 1998, the Ministry of Environment of the Republic of Korea designated, and officially announced, the largemouth bass as an invasive species, after conducting an ecological risk assessment. Even though the photoreceptor system plays a crucial role in invasion, very limited information is available on the photoreceptor cells of largemouth bass [8].

Most species of teleost fish possess color vision, and their retinal structure generally comprises of both rod cells for scotopic (low light detection) and cone cells for photopic

(distinguishes color) vision. However, several species exhibit high morphological variability, especially of the light-sensitive retina, which signifies adaptation to various ecological conditions and lifestyles [9–12], such as photic habitat types, flow velocity, feeding habits, and environmental conditions [13,14]. The vision generally depends on the size of the eye, density of photoreceptors, and number of retinal ganglion cells [15,16]. The retinas of diurnal fish have a higher abundance of cone cells for photosensitivity, whereas the retinas of nocturnal and deep-sea fish, which require scotopic sensitivity, are mainly rod-rich [15,17]. However, to the best of our knowledge, the relationship between the morphological structure of photoreceptor cells and the feeding behavior of largemouth bass has not been investigated yet.

Therefore, in the present study, the morphological characteristics and arrangement of photoreceptor cells of the largemouth bass were examined, with an emphasis on the relationships between environmental conditions and habitat. In addition, the prey-tracking ability of the largemouth bass was determined by analyzing and comparing its photoreceptor cell characteristics with those of *Coreoperca herzi* and *Lepomis macrochirus*, which are predatory fish with similar ecological traits [1–3].

Our evaluation of the correlation between the retinal structure and ecological habits of largemouth bass provides a basis for the production of fishing tools to catch invasive fish species, thereby enhancing their eradication.

## 2. Materials and Methods

### 2.1. Survey Site

This study was conducted in accordance with the Guide for the Care and Use of Laboratory Animals (2011), provided by the National Institutes of Health, USA. The protocol was approved by the Institutional Animal Care and Use Committee of Chonbuk National University. All surgeries were performed under MS-222 anesthesia, and all efforts were made to minimize pain. The largemouth bass subjects ($n = 10$) were collected from two sites, including the Mankyeonggang River, Deokjin-dong, Jeonju-si, Jeollabuk-do (35°50′ N; 127°07′ E), the Buckchosan Reservoir, Bodeok-ri, Daeya-myeon, Gunsan-si, and Jeollabuk-do (35°58′ N; 126°49′ E) in Korea during the non-spawning period. The lengths and weights of each of the collected largemouth bass were measured. This species is an invasive species in Korea and does not require a separate authorization request for collection.

### 2.2. Observation of Photoreceptor Cells

MS-222 (200 mg/L) was used as an able method of euthanasia of fish [18]. The eyes of the largemouth bass were extracted and fixed in a 10% buffered formalin for 12 h at 4 °C. Specimens were dehydrated with serially diluted ethanol solutions, ranging from absolute to 70%, cleared with pure xylene by sequentially replacing pure ethanol and xylene in a 1:1 ratio, and embedded in paraffin (Paraplast; Leica Camera AG, Wetzlar, Germany). Specimens were cut at 5 μm thickness (using a rotary microtome), deparaffinized, and stained with hematoxylin solution (Harris, Mayer's Modified) and eosin [19]. Light microscope (Carl zeiss-AX10, Jena, Germany) photographic and visual system evaluations were performed using the Carl Zeiss Axio Imager.A2 Vision LE Rel. 4.4 (Carl Zeiss AG, Jena, Germany) software.

The scanning electron microscope (SEM) specimens were prefixed in 2.5% buffered glutaraldehyde (0.1 M phosphate buffer, pH 7.2) [20]. The post-fixation was performed in 1.0% osmium tetroxide ($OsO_4$), prepared in 2.5% buffered glutaraldehyde. Once the specimens were dehydrated in a graded ethanol series (70–100%) and dried using a critical point method, the dried specimens were coated with gold ions. Thereafter, the specimens were observed using an S-450 SEM (S-450; Hitachi, Tokyo, Japan). For transmission electron microscopy (TEM), specimens were embedded in an Epon mixture (EMbed812; Electron Microscopy Sciences, Hatfield, PA, USA) [21], using the same methods of fixation and dehydration employed for SEM. The specimens were then observed using TEM (H-7650; Hitachi). To observe the gross morphology, serial semi-thin sections (0.5–1.0 μm thick)

were stained with 0.5% toluidine and observed under a light microscope. Both the radial and tangential sections were examined perpendicular and parallel to the retinal plane, respectively [14].

### 2.3. Data Analysis

The experimental values are expressed as a mean ± standard error of the mean. GraphPad Prism (version 9.0; GraphPad Inc., San Diego, CA, USA) one-way analysis of variance (ANOVA) was used for multiple comparisons, followed by Dunnett's test. Statistical significance was determined at $p < 0.05$.

## 3. Results

### 3.1. External Morphology of the Eye

The total lengths of the largemouth bass subjects, collected from sites 1 and 2, were 200–250 mm (mean 216 ± 15.2 and 228 ± 25.9 mm, respectively), and their weights were 150–200 g (mean 172 ± 19.7 and 179 ± 17.5 g, respectively). In addition, the collected largemouth bass had large eyes, with an average horizontal diameter of 25.2 ± 0.3% head length (both sites, Table 1). The eyes were clear, without eyelids, and had an elliptical shape with width greater than the length (Figure 1a,b).

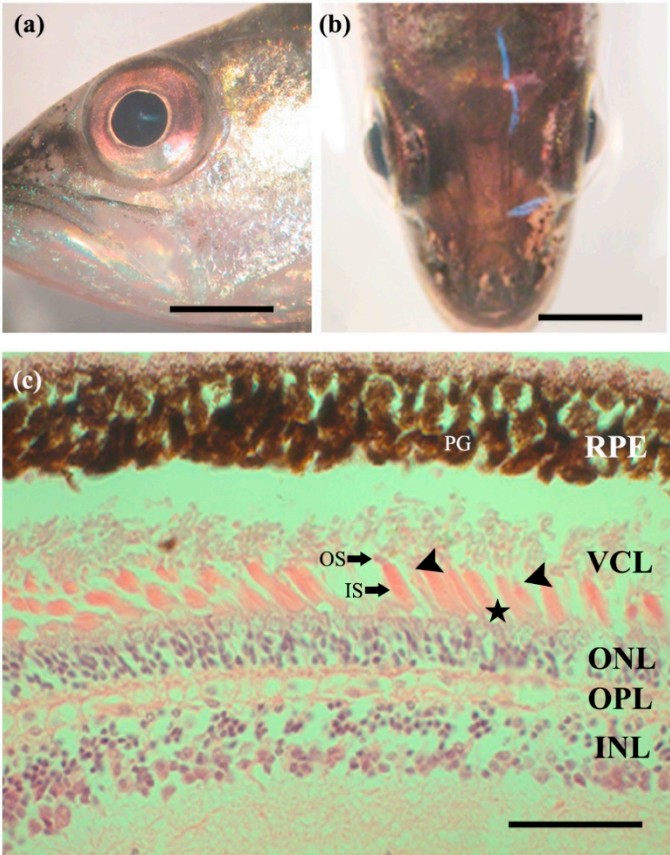

**Figure 1.** Characteristics of *Micropterus salmoides* eyes. (**a**,**b**) No apparent eyelids and clear, elliptical shape, with the horizontal length longer than the perpendicular length. Scale bars = 5 mm; (**c**) longitudinal section of the retina, composed of several layers. Single cone (arrowheads), double cone (asterisk), and rod cells, constituting the visual cell layer. Scale bar = 50 μm. INL, inner nuclear layer; IS, inner segment; ONL, outer nuclear; OPL, outer plexiform layer; OS, outer segment; PG, pigment cell; RPE, retinal pigment epithelium; VCL, visual cell layer; asterisk, double cone cells (equal); arrowheads, single cone cells.

**Table 1.** Eye diameter ratio, according to body length, weight, and head size of largemouth bass *Micropterus salmoides* (Pisces; Centropomidae).

| Site | N | Total Length (mm) | Weight (g) | Head Length (mm) | Eye Diameter (%) |
|------|---|-------------------|------------|------------------|------------------|
| 1 | 1 | 200 | 150 | 65.4 | 24.8 |
|   | 2 | 210 | 155 | 70.5 | 25.0 |
|   | 3 | 240 | 195 | 73.4 | 25.5 |
|   | 4 | 220 | 175 | 75.2 | 25.2 |
|   | 5 | 210 | 150 | 72.5 | 25.1 |
| 2 | 1 | 260 | 200 | 80.2 | 25.5 |
|   | 2 | 220 | 180 | 74.8 | 25.1 |
|   | 3 | 250 | 190 | 82.5 | 25.5 |
|   | 4 | 200 | 170 | 65.0 | 25.0 |
|   | 5 | 210 | 155 | 66.5 | 25.0 |

N: number of experimental fishes.

### 3.2. Retina

The radial section of light microscopy showed that the retina of the largemouth bass has a multi-layered structure. The layers, starting from the outermost layer to the layer closest to the vitreous body, are ordered as follows: the retinal pigment epithelial, visual cell, outer nuclear, outer plexiform, inner nuclear, inner plexiform, and ganglion cell layers (Figure 1c). The retinal pigment epithelium layer is the pigmented cell layer containing pigment grains or melanin granules that extends towards the outer photoreceptor segments, beneath a choroid layer filled with capillaries. The visual cell layer of the largemouth bass consists of densely arranged large rod cells, as well as single and double cone cells (Figures 1 and 2).

### 3.3. Photoreceptor Cells

Photoreceptor cells consist of two types of cone cells, including short single and equal double cone cells, along with bulky rod cells (Figure 2a–d). The single cone cells are acidophilic and have two distinctive segments: a small, conical outer segment and a larger, longer bulbous inner segment (Figure 2d,e). The single cones were, on average, $27.8 \pm 1.6$ μm in length and $3.82 \pm 0.2$ μm in diameter. The equal type double cone cells were eosinophilic and symmetrical, with similar lengths. Both elements of this cell type, known as twin (equal double) cone cells, were larger than the single cone cells, with an average length of $39.27 \pm 0.83$ μm and average diameter of $7.5 \pm 0.2$ μm. Similar to the single cone cells, the outer segment of these cells was short and cone shaped, whereas the inner segment was large and bulbous (Figure 2d). The cell extensions reached the outer plexiform layer (Figure 1c). The observation of single and double cone cells, through toluidine blue staining on semi-thin sections, revealed that the outer segment was weakly stained, but the inner segment was strongly positively stained (Figure 2c). The SEM results showed that the outer and inner segments were connected by calyceal processes (Figure 2d). In the tangential sections, the double and single cone cells were identified as a regular mosaic-like pattern of a petal arrangement, containing quadrilateral units composed of four equal type double cone cells surrounding a central short single cone cell (Figure 2a). As an anatomical unit, five single cone cells and 16 double cone cells were arranged per $25 \times 20$ μm area (Figure 2a,b).

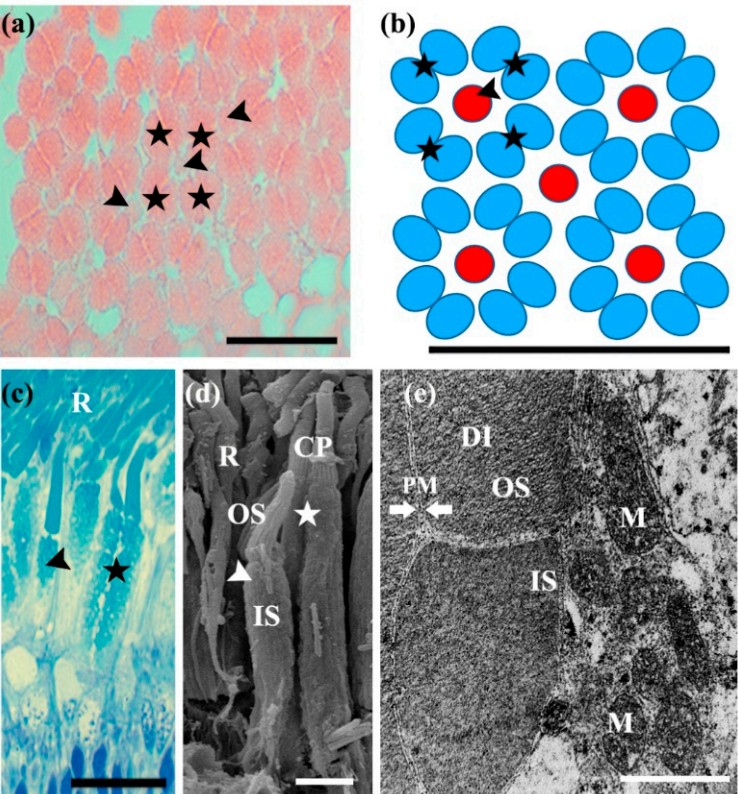

**Figure 2.** Morphological characteristics of *Micropterus salmoides* photoreceptor cells. (**a**) Transverse sections of the retina of an *M. salmoides*, observed under a light microscope. Scale bar = 20 μm; (**b**) mosaic pattern arrangement based on (**a**). The photoreceptor cell has two types of cone cells: equal type double (asterisks) and short single cone cells (arrowheads). As an anatomical unit, the mosaic model confirmed that 5 single (small red circles, arrowhead) and 16 double cone cells (large blue ellipses, asterisks) were arranged per 25 × 20 μm area. Scale bar = 20 μm; (**c**) longitudinal sections of photoreceptor cells, stained with toluidine blue and observed under a light microscope. *Micropterus salmoides* photoreceptor cells can be divided into three regions: single cone, double cone, and rod cell (R). Both rods and cones are divided into the inner segment (IS) and outer segments (OS). Scale bar = 20 μm; (**d**) scanning electron micrograph of photoreceptor cells: the outer segment links to an inner segment by calyceal processes (CP). Scale bar = 3 μm; (**e**) transmission electron micrograph of photoreceptor cells. The inner segment of photoreceptor cells has numerous mitochondria (M) and the outer segment has a multi-layer disc (DI) structure surrounded by plasma membranes (PM). Scale bar = 1 μm.

Large rod cells are typically composed of a single layer, a long, rod-shaped outer segment, and a small inner segment. The presence of the rod cells could be clearly identified in the semi-thin sections (Figure 2c,d). The rod cells were thinner and longer than the cone cells, with an average length of 75.6 ± 6.26 μm and average diameter of 2.3 ± 0.2 μm. The outer segment was positive for toluidine staining. The ellipsoid was also slightly positive and surrounded by multiple epithelial cells, whereas the myoid region was negative. SEM clearly identified that the calyceal process did not connect the outer and inner segments, and the inner segment consisted of separate ellipsoid and myoid regions (Figure 2d). The ultrastructure of the outer membrane was observed as a membrane disk stack, surrounded by a double membrane (Figure 2e), and the inner segments were rich in mitochondria. Even the same species were compared to identify differences in morphological characteristics of visual cells according to their habitat (river vs. reservoir). However, there was no morphological difference between the two sites.

## 4. Discussion

The retinas of most teleost fish contain photoreceptor cells, organized in a mosaic pattern, with cone cells as the dominant element, paired with randomly scattered rod cells [17,22]. The cone mosaic pattern is categorized into three types: row, square, and triangular. The row pattern includes a typical cone pattern, and has parallel-oriented double cones. In the square pattern, double cone cells are alternately perpendicular, with an angle of 60° or 90°. In the triangular pattern, double cone cells are predominantly arranged at angles of 60° and 120° [11,22–24]. A typical cone mosaic pattern of the square type is an arrangement of four double cone cells, surrounding a single cone cell [11,24]. Since the density and presence of various types of cone cells can influence the level of movement recognition in all directions, it is speculated that the arrangement, density, and diversity of cone cells are associated with high visual acuity [14,25]. Specifically, the density of three different types of cone cells and the presence of adequate rod cells may contribute to a rapid visual response to prey recognition. It can also enable long-distance migration and swimming [25]. The typical mosaic pattern, including double cone cells, may also be involved in the detection of polarized light, which is important in searching, as well as the recognition of prey in turbid waters [26]. This arrangement has been confirmed in shallow-water diurnal teleosts, whose survival is mostly sight-dependent [9,13,27,28]. We have previously reported the structure of photoreceptor cells in *C. herzi* and *L. macrochirus*, obtained through histological observation. As reported, these species are perciforms, similar to largemouth bass, and have the same composition of photoreceptor cells. In addition, they have similar mosaic patterns. *C. herzi* and *L. macrochirus* are both predatory [14], but differ in their habitat and feeding depths. We observed a difference in the diameters of the three types of photoreceptor cells between these two species and largemouth bass (Table 2 and Figure 3). In largemouth bass, the diameter of photoreceptor cells was smaller than that in *C. herzi* and similar to that of *Lepomis macrochirus*. The invasive species, largemouth bass and *L. macrochirus*, consisted of cells similar to those of *C. herzi*, which is highly related to their habitat. Unlike the Largemouth bass, the *C. herzi* hides in the crevices of the rocks at the bottom of the river and only attacks when the prey comes in front of the micro-habitat. In our previously reported results, it was confirmed that *C. herzi* has colored cornea, as a result of having adapted to the dark environment [14]. These results show a similar photoreceptor cell composition and arrangement but *C. herzi* inhabits the bottom, and alien species live in the middle and upper layers, so it can be explained that the visual cells reflect their habitat well. Furthermore, as the diameter of the photoreceptor cells was small, the density of cells in the cone-mosaic pattern was high, which is likely to be closely related to the hunting ability of the largemouth bass. Hunt et al. [29] also reported that the square mosaic arrangement of cone cells is specialized for moving objects and prey, and the results of adaptation in favor of predation of living fish are well reflected in eyes and visual cells [29]. Overall, the composition, arrangement, and density of largemouth bass photoreceptor cells are related to sensitive visual acuity, an important factor in determining the feeding area within the habitat.

**Table 2.** Photoreceptor cell diameters of three species of predatory fish (*Coreoperca herzi*, *Lepomis macrochirus*, and *Micropterus salmoides*).

| Cell Types | (μm) | *Coreoperca herzi* | | *Lepomis macrochirus* | | *Micropterus salmoides* | |
|---|---|---|---|---|---|---|---|
| | | Mean ± SD | Range | Mean ± SD | Range | Mean ± SD | Range |
| Rods | Length | 75.3 ± 6.2 | 69.3–83. 8 | 74.7 ± 5.7 | 69.6–83.3 | 75.6 ± 6.3 | 69.6–83.3 |
| | Diameter | 3.1 ± 0.24 | 2.9–3.33 | 2.7 ± 0.2 | 2.52–2.9 | 2.3 ± 0.2 | 2.2–2.7 |
| Single Cones | Length | 24.0 ± 0.04 | 23.6–24.9 | 31.4 ± 3.0 | 25.0–32.4 | 27.8 ± 1.6 | 25.0–32.4 |
| | Diameter | 6.6 ± 0.5 | 6.0–7.3 | 3.3 ± 0.3 | 2.9–3.7 | 3.8 ± 0.2 | 3.3–4.1 |
| Double Cones | Length | 21.8 ± 1.3 | 20.0–25.1 | 34.3 ± 2.1 | 38.7–40.6 | 39.3 ± 0.8 | 38.7–40.6 |
| | Diameter | 11.3 ± 0.4 | 10.6–12.2 | 7.5 ± 0.4 | 7.1–8.2 | 7.5 ± 0.2 | 7.46–7.9 |

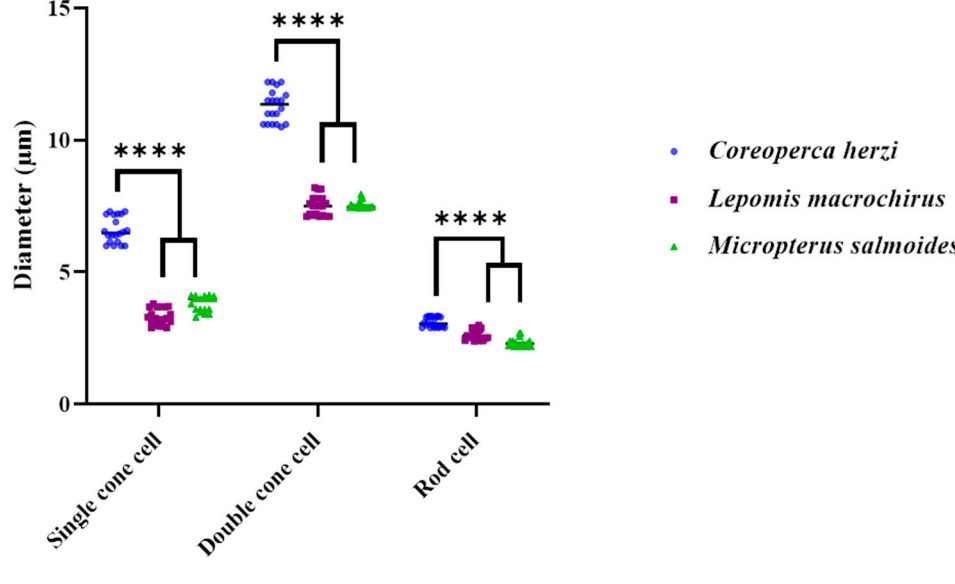

**Figure 3.** Comparative analysis of photoreceptor cell diameters between three species of predatory fish (*Coreoperca herzi*, *Lepomis macrochirus*, and *Micropterus salmoides*). Values are expressed as means ± standard error of the mean. **** $p < 0.0001$, when compared with the photoreceptor cell diameter of *C. herzi*.

Mitchem et al. [8] modeled a simulation of the color vision and behavioral analysis of the largemouth bass in Florida and Illinois (US), using a visual model that detects and distinguishes different colors under varying light conditions. This study measured the spectrum of photoreceptor cells of nine fish to identify rods, single cones, and twin cones and observed that single cones were sensitive to green, while twin cones were sensitive to red. Additionally, no significant differences among the three different types of photoreceptor cells were observed in either Florida or Illinois [8]. The morphological structure of the photoreceptor cells of the largemouth bass observed in our study was consistent with these observations. Notably, their study conducted a behavioral analysis using olfactory cues to determine that, although the largemouth bass cannot usually distinguish yellow, it could distinguish the color with poor accuracy upon training. Based on the findings of this study, largemouth bass were assumed to survive rapid and gradual habitat changes.

The largemouth bass has displayed a wide feeding range, under varying environmental conditions, and significant adaptability and reproducibility in the natural environment of Korea over several years. According to our findings, as well as those of Mitchem et al. [8], these characteristics can be attributed to their sensitive visual detection ability, resulting from the highly developed photoreceptor cell structures and high distribution rate of the three types of photoreceptor cells. These characteristics may have led to the rapid occupation of freshwater ecosystems and outcompeting indigenous fishes in Korea.

## 5. Conclusions

The largemouth bass (*M. salmoides*) is an invasive species in Korea. Although originally imported as a food source, it has now become a top predator in Korea's river ecosystems and impacts native fish and invertebrate populations. Therefore, the habitat and behavioral adaptations of largemouth bass were examined by studying its visual system at the microscopic level. The largemouth bass photoreceptor cells consist of two types of cone cells (short type single and equal types of double cone cells), as well as long, bulky rod cells. All photoreceptor cells can be divided into inner and outer segments; the inner segments were rich in mitochondria, and the outer segments had a stacked disc structure.

The largemouth bass has three photoreceptor cells that allow it to visualize the prey in deep water, exhibiting the visual characteristics of typical predatory fish. However, unlike similar predatory fish, the photoreceptor cells were small in diameter and high in density. Other studies posit that largemouth bass may be able to adapt to color. In summary, these results suggest that largemouth bass, an invasive species, can inhabit numerous habitats to become a dominant freshwater species in Korea. This correlation between the feeding habits and visual characteristics of the fish provides crucial data for studies on the ecology of freshwater fish in Korea. Furthermore, our study provides a basis for making tools to capture invasive species by understanding the ecological habits of invasive species.

**Author Contributions:** Conceptualization, J.G.K. and S.-H.Y.; data curation, J.G.K. and S.-H.Y.; formal analysis, J.G.K. and S.H.K.; funding acquisition, J.G.K.; investigation, J.G.K. and S.H.K.; methodology, J.G.K.; project administration, J.G.K. and J.Y.P.; resources, J.G.K. and S.H.K.; software, J.G.K. and S.-H.Y.; supervision, J.Y.P. and S.-H.Y.; validation, J.G.K., S.H.K., J.Y.P. and S.-H.Y.; visualization, J.G.K.; writing—original draft, J.G.K. and S.-H.Y.; writing—review & editing, J.G.K. and S.-H.Y. All authors have read and agreed to the published version of the manuscript.

**Funding:** This research was supported by the Basic Science Research Program through the National Research Foundation of Korea (NRF), which is funded by the Ministry of Education, Science, and Technology (NRF-2017R1A6A3A01077013).

**Institutional Review Board Statement:** This study was conducted in accordance with the Guide for the Care and Use of Laboratory Animals (2011), from the National Institutes of Health, USA. The protocol was approved by the Institutional Animal Care and Use Committee of Chonbuk National University. All surgery was performed under MS-222 anesthesia, and all efforts were made to minimize pain.

**Informed Consent Statement:** Not applicable.

**Data Availability Statement:** The datasets generated during and/or analyzed during the current study are available in Appl. Microsc. 2016;46:89–92 and J. Appl. Ichthyol. 2014;30:172–174, https://doi.org/10.9729/AM.2016.46.2.89 (accessed on 1 January 2022) and http://dx.doi.org/10.1111/jai.12311 (accessed on 1 January 2022), respectively.

**Acknowledgments:** The authors acknowledge all researchers affiliated with the Alpha Ecology Institute for their co-operation in facilitating sample collection. We thank Cheol Woo Park, Yun Jeong Cho, and Jong Wook Kim, who helped in the sample collection for this work. We also thank all anonymous reviewers, whose thorough and thoughtful reviews greatly improved this manuscript.

**Conflicts of Interest:** The authors declare no conflict of interest.

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
