# Peer review of "Correlation between Feeding Behaviors and Retinal Photoreceptor Cells of Largemouth Bass, Micropterus salmoides, in Korea"

_fishes, doi:10.3390/fishes7010025_

Round 1

Reviewer 1 Report

Line 2-4:  In fact, the feeding habitats of the largemouth bass were not specifically described in the text.

Line 33-35:  Since the largemouth bass possessed highly developed lateral line and lightning-fast speed, how to ascribe the feeding habitats solely to the vision, or photoreceptor cells?

Line 35-36:  Although the largemouth bass is diurnal carnivore that hunts other fish, it is most active in low light during the morning and evening [12]. Whether and how your results support this opinion/conclusion?

Line 58-61:  The photoreceptor cell characteristics of the largemouth bass were compared with those of Coreoperca herzi and Lepomis macrochirus, which are predatory fish with similar ecological traits. How to define their similarity or difference between them?

Line 71-74:  There were two fish groups were sampled from rivers and reservoirs, however, your data didn’t show whether there were some differences in photoreceptor cells detected from the different environments (river vs. reservoir)? Meantime, I also wonder whether there are some differences in feeding habitats between river and reservoir living stocks.

Tab 2 and Fig 3:  The photoreceptor cells, rod, single, and double cone cells of M. salmoides were smaller (2.3 ± 0.2, 3.82 ± 0.2, and 7.5 ± 0.2 μm, respectively) than those of other species 23(3.1 ± 0.24, 6.6 ± 0.5, and 11.3 ± 0.4 μm in diameter, respectively in C. herzi). How to explain their size difference in photoreceptor cells and correlation to their feeding habitats?

Author Response

Thanks for the review and comments. This is Su-Hyang Yoo from National Institute of Ecology, Korea. This is the response letter for the revision of the manuscript titled as “Correlation between Feeding Behaviors and Retinal Photoreceptor Cells of Largemouth Bass, Micropterus salmoides, in Korea” (fishes1507603). 

Reviewer 2 Report

Introduction:

Line 45-46: why are there underlined sentences?

Materials and Methods:

Line 81-82: If you dehydrate you go from 70% of ethanol to the absolute one. How many time do you left the samples in ethanol and xylene?

 What software did you use to analyse the data?

Results:

Line 107: "The total lengths..." instead "The toral lengths...".

Table 1: N is the size of the sample or is a number assigned to each fish? Maybe you can specify it in the table legend. 

Figure legend 1: "RPE, retinal pigment epithelium" instead of "RPE, retina pigment epithelium".

Suggestion to figure legend 1: "Single cone (arrowheads), double cone (asterisk)..." instead adding the meaning of the labels at the end. I think that it is easier when you read to see at the same time what you want to point at. 

Figure 1: Why is there a green background? Can it be improved this image?

Figure 2: in image a the background still green. 

Figure legend 2: (c) "longitudinal sections of photoreceptor cells stained with toluidine blue..." instead of "longitudinal sections of photoreceptor cells stained with toluidine". Why do you use the abbreviation of light microscope on (c) but not in (a)? You don't say how many um are the scale bars of a and b.

Line 152: how can I distinguish the segments in figure 2a? Maybe you should point only to figure 2c,d.

Line 156-157: there is an enter.

Discussion:

Suggestion: describe the comparison of the three fishes to results (figure 3 and table 2) and discuss the differences in the discussion section.

Figure 3: What software did you use to create the chart? Maybe you can resize the dots, triangles and squares (smaller).

Author Response

(The authors gave the same response as above.)
